# Making a Case for the Inclusion of People with Intellectual Disabilities in Higher Education

**Deirdre Corby** [1,*], **Eilish King** [2], **Mary Petrie** [1], **Schira Reddy** [1], **Aisling Callan** [1] and **Toff Andersen** [1]

1 School of Nursing, Psychotherapy and Community Health, Faculty of Science and Health, Glasnevin Campus, Dublin City University, D09 V209 Dublin, Ireland; mary.petrie@dcu.ie (M.P.); schira.reddy@dcu.ie (S.R.); aisling.callan8@mail.dcu.ie (A.C.); tandersen@pobal.ie (T.A.)

2 Discipline of Occupational Therapy, Trinity College Dublin, D08 W9RT Dublin, Ireland; kinge1@tcd.ie

* Correspondence: deirdre.corby@dcu.ie; Tel.: +353-17008524

**Abstract:** The 21st century to this point has seen increased diversity throughout the student population in higher education. Many stakeholders value this diversity as it enhances the overall education experience for all students. While the number of students with disabilities in higher education is rising, challenges and barriers to participation remain, including for people with intellectual disabilities. The intent of this paper is to make a case for the inclusion of people with intellectual disabilities in higher level education. Following the introduction to the topic, the existing literature on inclusion in higher education is explored, and issues such as challenges to inclusion are discussed. We then explain the situation in the Republic of Ireland, detailing how one university-based programme, the Dublin City University (DCU) Ability project, aims to prepare people with intellectual disabilities to move towards employment. The results of a survey sent to the staff in the University ($n$ = 112), exploring their knowledge and views of the project, is then presented. While the project has been well received by learners and their supporters, it was important to identify the views of University staff to promote the future growth and sustainability of the project. The common themes identified were inclusion and diversity, communication issues, and positioning the project in the University's Strategic Plan. The results contribute to the case for inclusion as university staff indicate their support while acknowledging challenges. These challenges are primarily in the area of communication but also in ensuring that university goals are implemented to provide a diverse and inclusive university community. We make the case that diversity should be valued, and that universities should be a space for all students, including those with intellectual disabilities, to learn and thrive.

**Keywords:** inclusion; intellectual disabilities; higher education; academic staff





## 1. Introduction

In the Republic of Ireland, children who have special needs either attend special schools, or mainstream schools in special classes or with supports. Once their primary and secondary education is complete at the age of eighteen, those with intellectual disabilities tend to progress to specialist services rather than higher education. Recently however, diversity in education is being valued more as students from socio-economically disadvantaged backgrounds, mature students, members of the travelling community, students with disabilities and international students are now being included in higher education. This includes over 11,000 students with disabilities, representing 5.2% of the entire student population [1]. The Disability Access Route to Education (DARE) in the Republic of Ireland, includes those with autistic spectrum disorder, attention deficit disorder/attention deficit hyperactivity disorder, blind/vision impaired, deaf/hard of hearing, developmental co-ordination disorder (DCD; including dyspraxia), mental health condition, neurological conditions, speech & language communication disorder, significant ongoing illness, physical disability and specific learning difficulty, but not those with intellectual disabilities.

There are, however, a variety of programmes available across a number of Colleges in the Republic of Ireland for students with intellectual disabilities [2]. These programmes utilise different models of inclusion and supports [3]. In 2019, the Inclusive National Higher Education Forum (INHEF) was created to support:

> " ... *the sustainability and development of existing and future inclusive education initiatives for students with intellectual disabilities within Irish Higher Education Providers*" [4]

There are similar programs offered internationally with examples including the Canadian On Campus project at the University of Alberta, which has been running for over 30 years. In the USA, Think College works with 259 Colleges that offer a variety of programmes supporting people with disabilities. Flinders University in Adelaide, Australia has been offering the Up the Hill project for over 20 years. The University of Sydney hosts the Uni 2 Beyond programme for adults with intellectual disabilities.

Dublin City University (DCU) has been offering an employment-focused programme called the DCU Ability project since 2018 for young people with disabilities. Based in the School of Nursing, Psychotherapy and Community Health, a School within the DCU Faculty of Science and Health, it aims to support people with intellectual disabilities in building the skills and confidence they need to progress to further education or towards the workplace. Learners were recruited through services for people with disabilities and initially open evenings were held to inform learners, families and service staff about the project. Potential learners would meet with the Ability team to ensure the course would meet their needs and help them move towards employment. It was one of 27 projects co-financed by the Irish Government and the European Social Fund as part of the ESF Programme for Employability, Inclusion and Learning 2014–2020. The project was extended to December 2022 with additional funding from the Irish Government via the Dormant Accounts Fund. It was important to explore the impact of the project among university staff in particular, to support future development and promote sustainability within the university. While studies have found that inclusive approaches are seen as positive for all stakeholders [5], and that academic staff have a significant role to play in ensuring students with disabilities feel included and welcomed [6], challenges to inclusion still exist. Inclusion in higher education is explored further, and challenges are discussed, prior to explaining the DCU Ability project and the results of a survey conducted with university staff.

## 2. Inclusion in Higher Education

Inclusion in higher education for people with intellectual disabilities is a growing area of interest in research and practice. Uditsky & Hughson [7] espoused inclusive education as an evidence-based moral imperative, arguing that inclusive practices benefit everyone. They also highlighted the moral obligation to support students with intellectual disabilities to have opportunities in higher education, in order to avoid the vulnerabilities associated with segregation. In the USA, Grigal & Hart [8] with others have investigated how inclusion has evolved, the supports required by institutions to meet challenges and minimise exclusion [9], the importance of transition planning and employment [10] and the different options available across the USA [11]. More recently, McCabe et al. [12] have reported on their study with faculty members who related the many positive effects for both faculty and students when students with intellectual disabilities are included in mainstream courses.

In Canada, Bruce Uditsky has been recognised for his work in advocating for inclusion and has also published many significant papers. He believes in advocating for an authentic student experience for students with intellectual disabilities, and also highlights the valuable contribution all students can make to society [13]. The experiences of people with intellectual disabilities have also been researched and discussed by these authors and many others. Students with intellectual disabilities report a range of benefits of inclusion such as improved confidence, independence and increased social networks [14], while acknowledging that assessments can be stressful [15]. They also experience higher education

as creating opportunities to build on existing skills and achieve unexpected goals [6], while " . . . *engaging in balanced, meaningful relationships during the college experience that are not encountered by their . . . peers*" [16] (p. 58). Other recent research has shown that attending higher education can lead to more employment opportunities; in particular for women with disabilities [17], and that a link to employment is essential for career progression [18].

In making the case, there are numerous arguments offered regarding the benefits of inclusion in higher education for people with intellectual disabilities; additionally, there is the impact on social change and on others, such as fellow students, lecturers, family and friends. Society benefits, as people have greater opportunities for employment and earning prospects [16,19–22]. Students without disabilities have identified how much they value the opportunity to engage with a diverse student body [5,12,23–25], while all students benefit from a richer university experience [16]. Lecturers highlight the need to raise expectations among students and the benefits of non-disabled students interacting with students with disabilities [15]. Families play significant roles in the lives of people with intellectual disabilities and want to see their opportunities increase [6,26,27], and can provide support for both transition and subsequent inclusion in higher education [28]. Overall however, this quote from Uditsky and Hughson highlights the contribution inclusion makes for all in higher education, as it " . . . *clearly contributes to drawing out the best in students with intellectual disability, their peers and faculty*" [7] (p. 301).

## 3. Challenges for Inclusion

Challenges for inclusion still exist, despite many successes and reasons to be positive. In the Republic of Ireland, as in many other countries, there is evidence of some Higher Education Institutions discontinuing programmes or scaling back their offerings. Challenges such as attitudinal and structural barriers have been highlighted [15,16,18], with higher educational institutions needing to fully understand inclusion to ensure meaningful opportunities for students with intellectual disabilities [6]. An Irish report found that students received poor or limited career guidance, and that there are barriers to successful transition and limited post-school options [29]. Young people themselves reported challenges in transitioning from education to work, with problems in areas such as work placements, work readiness, securing and then retaining employment [30], in addition to low expectations of their capabilities [31]. There is a link between education and employment, and evidence that progression to employment is not seen as a priority for those with intellectual disabilities [29]. One significant challenge is related to funding to support people to attend [16], with many projects (including DCU Ability) being time-limited due to funding streams.

## 4. The Irish Context and the DCU Ability Project

In the Republic of Ireland, the current Minister for Further and Higher Education, Simon Harris, has acknowledged the importance of both further and higher education and employment for people with intellectual disabilities. This has led to an updated national access plan, completed in May 2021, aiming to support students' access higher education once they have completed secondary school [32]. Similar to previous plans, the new legislation aims to increase representation from groups that have traditionally been excluded from higher education. The government is investing in inclusive education programmes in some universities in Ireland.

The DCU Ability project (2018–2022) is focused on employment by providing education, training and supports for young adults with intellectual disabilities to explore their opportunities for work. By May 2022, the project had supported 150 young adults, 41 of whom attended a campus-based course, 98 completed an on-line blended course while 11 learners availed themselves of one-to-one supports.

While the project aims to support the learners to become career-ready, it also gives them an opportunity to attend university and experience aspects of life on campus. Occupational therapy and career advice are integrated into the campus-based and online courses. Closure

of the university campus during the COVID-19 pandemic necessitated cessation of the campus-based course and work experience placements. The team designed and delivered a new blended learning course for young adults with intellectual disabilities during this time. The move to on-line resulted in increased participation from students across the country and allowed for welcome engagement at a time when services had to be restricted for people with intellectual disabilities.

Feedback and consultation with key stakeholders (including learners) were important aspects of the project evaluation. University staff in particular play a vital role in promoting inclusion, either directly by offering work placement, or indirectly by fostering a welcoming and inclusive atmosphere on campus. Staff can affect positive attitudes [6,12,33] and identify supports required [15,18]. They can encourage other students to include students with disability in all aspects of university life [34]. Thus, it was important to gather the experiences and views of university staff in relation to the DCU Ability project. This has the potential to inform future project planning and identify strategies to enhance the inclusion of students with intellectual disabilities on the university campus.

## 5. Materials and Methods

To elicit the knowledge and views of DCU staff, data were collected through an online survey tool designed by the research team. Participants were asked to respond on a Likert scale to 10 statements regarding the DCU Ability project to elicit their knowledge and understanding of the project. There was an opportunity then to give feedback on their overall views of the strengths and challenges of the project. Participants included all University staff who have access to the "DCU all staff" email. Therefore a wide range of academic, technical and professional staff would have received the email with the survey link. It is most likely that the majority of those who responded to the survey had direct experience of working with the learners. There are over 3000 staff and research personnel eligible to receive emails. It is also possible that due to the high volume of emails they receive, a response rate of just below 4% ($n$ = 112) was worthy of investigation. The questionnaire was distributed during a period of university campus closure due to COVID-19 pandemic restrictions, which may have had a further adverse impact on the response rate. Quantitative responses were analysed descriptively while qualitative data were analysed thematically.

Ethical approval for this study was obtained from Dublin City University, Research Ethics Committee. Participation was anonymous, with no demographic or identifying information collected. After the initial email asking people to answer the online questionnaire, one reminder email was sent.

## 6. Results

Quantitative data are presented using a stacked bar graph describing participant responses to ten statements. Participants were asked to select from 1 to 7 on a Likert scale with 1 = strongly disagree and 7 = strongly agree. Thematic analysis was applied to the qualitative data resulting in three themes: inclusion and diversity, communication issues and positioning the project in the DCU Strategic Plan.

### 6.1. Quantitative Data

Participants were presented with ten statements regarding the DCU Ability project, and their understanding of the project itself and its impact on the University. Figure 1 gives a graphical representation of their responses ranging from strongly agree to strongly disagree.

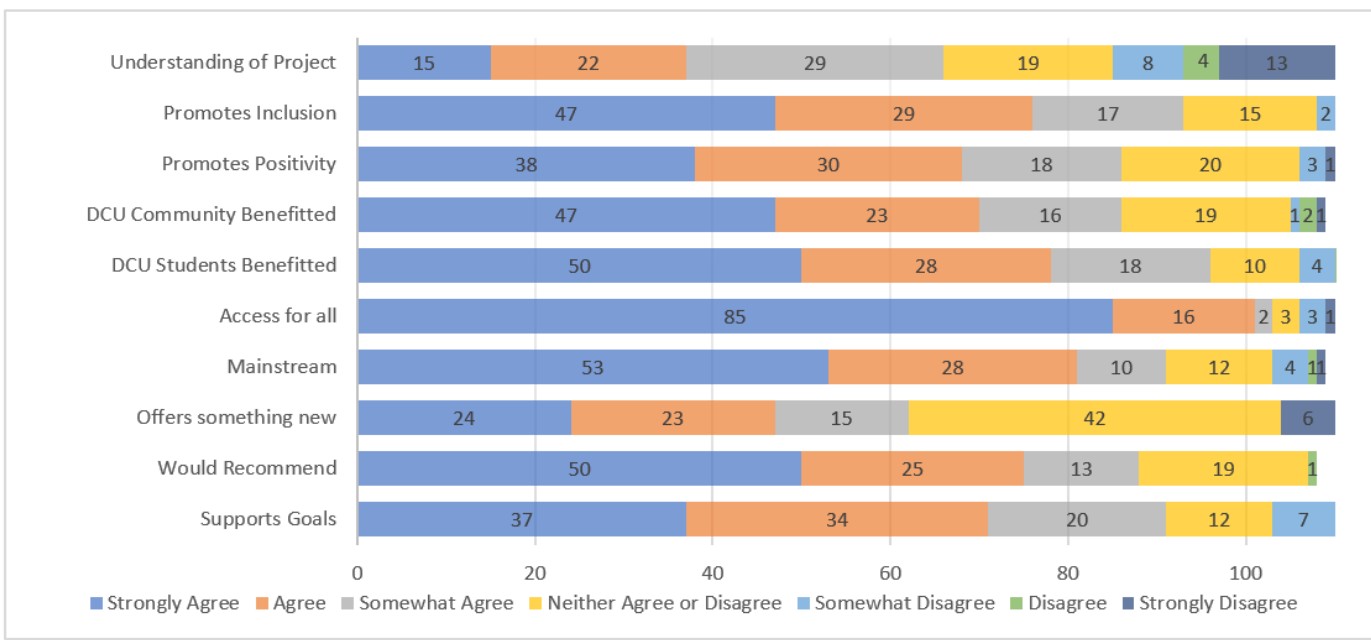

**Figure 1.** Survey Reponses using Likert scales 1–7.

We asked participants if they understood what the DCU Ability project is trying to achieve. This elicited a mixed response—just under 14% (*n* = 15) of participants said they strongly agreed, while most responses were between 6 and 4 on the scale; 64% (*n* = 70) indicated that people may be unsure about what we are trying to achieve, and a further 15% (*n* = 25) are very unsure or do not know what we are trying to achieve.

Participants then responded to the statement "*I believe that DCU Ability allows DCU to be more inclusive for people with disabilities*". Responses show that most people believe the project promotes inclusion, with over 69% (*n* = 79) indicating a 6 or 7 on the scale.

In a similar way, the next statement showed positive results in the response to the statement "*Knowing about the DCU Ability project has made me think more positively about the DCU Community*". Here nearly 62% (*n* = 68) responded in the 6–7 range with only one participant strongly disagreeing with the statement.

The statement "*I believe the DCU Community benefited from working with the young adults on the DCU Ability project*", again shows positive results, with over 64% (*n* = 70) choosing between 6 and 7 on the scale, and 32% (*n* = 35) choosing between 4 and 5.

Participants responded very positively to the statement "*I believe DCU students benefit from the DCU Ability students having access to DCU campuses*". A majority 86.5% (*n* = 98) agreed or strongly agreed, showing that participants could see benefits for mainstream students when students with disabilities are included on campus.

Again a very clear result was found in the statement "*I believe people, regardless of their ability should have access to DCU*", with 77% (*n* = 85) people strongly agreeing with this statement.

When asked to respond to the statement "*I believe that the DCU Ability project should become part of mainstream DCU programmes*" most responders strongly agreed, with over 76% (*n* = 81) agreeing strongly or very strongly.

Of significance is the result to the statement "*I believe the DCU Ability programme offers something which is not available in other universities or higher education institutes*". While 43% (*n* = 47) agreed or strongly agreed, a significant number 38% (*n* = 42) chose the mid-range of 4 on the scale. This may indicate a lack of knowledge of the details of the programme itself.

Responses to the statements overall indicate very good support for the project, so when asked to respond to "*I would recommend the DCU Ability project to others*", 69% (*n* = 75) agreed or strongly agreed.

The final statement was "*I can see how the DCU Ability project supports DCU's strategic goals*". The majority 64.5% (*n* = 71) agreed or strongly agreed with the statement, with no participants strongly disagreeing with the statement.

On the graph above, the strong response from the DCU Community regarding access for all is evident. It also shows the need for better promotion of the project, as while most responses are positive, there is evidence of a lack of understanding of the project. This is discussed further in the analysis of the qualitative data.

*6.2. Qualitative Data*

We offered participants an opportunity to add qualitative comments regarding any benefits they think DCU has gained from the project and any aspects/issues they felt could have been improved. Using thematic analysis, we identified three themes from the data: inclusion and diversity, communication issues and positioning the project in the DCU Strategic Plan. Details from the quantitative data are used to enhance these further findings.

6.2.1. Inclusion and Diversity

Inclusion and diversity emerged as the main theme from the comments received from participants. Many wrote briefly about the need for "*more inclusion*" or "*diversity and inclusion*" and "*a more diverse population, or student body*". One participant stated that the DCU Ability project gives "*A message to communities that DCU is a university for all. A more diverse environment*". The project has impacted both on staff and students as they could see the benefits for everyone in the University of having a more diverse student body included in everyday life in DCU. One participant commented, "*It has become a more inclusive campus which has also gained widespread recognition for being so. Furthermore, this encourages the academic staff to both recognise and cater for students with different levels of ability etc.*" Another staff member remarked "*I believe that it has made DCU more inclusive and given people who might not ordinarily have come into contact with DCU Ability participants the ability to do so and to see what these young adults have to offer*". One comment summarises the experience of staff, saying "*Increased participation and inclusion from young adults with disabilities can only be a good thing*".

Overall, the results provide a picture of how staff in DCU perceive the inclusion of people with intellectual disabilities in higher education. A total of 43% (*n* = 47) of participants strongly agreed that they benefited from working with young people with disabilities. Specifically, staff observed a more inclusive and friendly campus upon encounters with the students. This finding is typified by one comment of staff members who said

> "*We (academic staff) benefited with gaining a new understanding of disabilities and their abilities in offering them an inclusive educational experience*", and "*I enjoyed interacting with the ability students and seeing their enthusiasm for college life from a social and academic perspective. They bounced into DCU in the mornings with happiness and smiles for everyone, the catering staff in particular in Nursing building loved having them in their cafe as they were always happy, independent and appreciative of college life*".

The participants of this study support inclusion and want to promote diversity in higher education. A total of 92% (*n* = 101) of participants either very strongly or strongly agreed that students with disabilities should have the right to access the educational system regardless of their disability. The majority of participants also agreed with mainstreaming such projects and would recommend them to others.

One participant went further, explaining how the experience of having students with disability impacted positively in their department.

> "*My department has hosted two DCU Ability learners since the project started, and, strictly as an aside, the days the learners were there were always the happiest in the office. I regret that the pandemic has put the programme on ice for the moment, because I think we were beginning to learn a great deal about how accessible our service really is (or, more to the point, isn't)*". These benefits extended to university departments revisiting if their

spaces were accessible and inclusive. *"It gave us the opportunity to reassess how we work, the environment that the work happens in, and the business processes that we use that make perfect sense from inside the department, but which really lack for efficiency and transparency seen through a fresh pair of eyes"*.

Participants articulated the many benefits of inclusion and diversity, but also included a challenge to the university reflected in this contribution:

*"DCU Ability project challenges our idea of who campuses, university facilities and higher education are for. This is hugely impactful. I feel it challenges our narrow belief of what academic or professional achievement looks like and who can achieve it. People with intellectual disabilities are absent from our campuses, and DCU Ability in a small way broke this invisibility. DCU Ability should continue—DCU has the resources that can assist the personal, academic and professional development of young people with IDs, can provide meaningful training and employment opportunities. Everyone benefits from inclusive and diverse environments, and DCU has a role to play"*.

Others referred to the need to raise awareness for the university to be a place for all, to recognise the barriers to inclusion some people with disabilities face, and the need for initiatives such as the DCU Ability project to break down barriers. However, they acknowledged the challenges: " ... *the work is very important from an inclusion and diversity perspective. I imagine it is very hard to implement in practical terms, but please continue to develop it"*.

Overall the contributions from staff show that they value inclusion and diversity and see the benefits for staff and students but also see the complexities of inclusion and this is evident in the next theme, which highlights the many issues around communication.

### 6.2.2. Communication Issues

Communication issues have been indicated, as 40% (*n* = 44) of participants rated their understanding of the project between 4 and 1 on the Likert scale. When asked about improving areas or aspects of the project, one participant remarked *"I don't think the general DCU community had enough awareness or exposure to the DCU Ability students to have had a large impact at the wider level"*. Another participant reported *"I do not believe that as many people that should be aware of the project are. I suggest that the more people are aware and witness the project the better as it creates more understanding overall"*.

Participants reinforced the difficulty they encountered to learn about the project through academic communication channels, including email. One faculty member commented *"but there are a lot of DCU staff communication emails and hard to understand everything when there is a lot of work on constantly"*. Similarly, another participant remarked *"There is so many different press-releases and announcements, some of which appear to emit more heat than light, and the result is that this particular project was not on my radar, unfortunately"*. Some went further, saying *"I don't know much about it, how come?"*, *"Had not heard of DCU Ability before this email"* and *"I didn't know it existed"*.

Clearly, there are difficulties with knowledge and awareness of the project across the university, but staff embraced this problem, making numerous suggestions as to how communication could be improved to increase levels of visibility for the project. Staff called for initiatives aimed directly at academic staff. For example, one participant suggested *"More meet and greet sessions with DCU staff at all levels within our DCU community"*. Staff in the survey also cited training opportunities to educate DCU faculty members. Another commented *"While I was aware of the project and team members, I was not aware of the rules and functions"*. The above comment ties in with another participant's suggestion to introduce *"dedicated resources for staff to support the initiative"*. Similarly, a member of staff beginning their career in DCU remarked on the need to offer induction training related to current inclusion projects: *"As a new member of staff, I know very little about the DCU Ability project, so perhaps this could be part of our induction. I would be very interested to find out more"*. Other suggestions by participants sought to integrate disabled students further into campus life.

One lecturer observed "*I would love to have been given specific ways how I can participate in the project, for example, by including participants in the DCU Ability project in my classes as teaching aids, assistants, learners, listeners etc.*" Similarly, a staff member advised "*mentoring projects with the wider student body so that all students are mingling*" as well as "*inclusion in clubs and society*", pointing to how students with disability may enjoy the social aspects of attending college.

While communication issues have impacted the knowledge and understanding of the project among university staff, their comments also offer many ways to improve communication and promote inclusion. To move towards a more inclusive and diverse university, having strong alignment with the university's strategic goals would be very positive. In the final theme, this link with the goals was emerging from the qualitative data.

6.2.3. DCU Strategic Goals

Results indicated that surveyed staff felt that the project is congruent with DCU's strategic goals. A total of 64.5% (*n* = 71) of participants strongly agreed or agreed with the project's alignment to the university's strategic goals. There is evidence that staff perceive inclusion as an explicit value for their university. Here, employees referred to the DCU Strategic Plan 2017–2022, which offers an extensive five-year map to support its student and staff body and the wider society. What is particularly relevant is DCU's goal to "*give everyone the chance to shine*" by realising every student's potential, including those with additional needs. This is reflected by one participant's comment: "*This is a great initiative that has increased awareness within DCU about the educational and training needs of people with disabilities. It has delivered in a meaningful way on the university's aim to make DCU an inclusive place where everyone has the chance to shine*". Others refer to the project as " . . . *putting into practice the university's communal ethos*", "*looking beyond the traditional role of education, broader perspective and working for the greater good*" and " . . . *a chance for DCU to practice what it preaches. A chance for all staff and students to see first-hand the systemic and institutional barriers that this community face . . . a chance to be part of the solution with a better understanding of the broad concepts of UDL [Universal Design for Learning], access, transfer and inclusion*".

These results show that staff are aware of the university's goals, but also recognise that there can be challenges to reach those goals and projects like DCU Ability can support their achievement. "*An understanding of the need to allow the university to be a place for ALL regardless of societal norms*".

On a central level, values of inclusion are reflected in DCU's various diversity initiatives (under Strategic Goal 5), including projects such as the Age-Friendly University and Women in Leadership initiatives. In line with their five-year strategy to prepare all students for the rapidly evolving Irish workplace, the DCU Ability project supports young adults with disabilities in their career paths. Remarking on the project's role in DCU, one participant commented:

> "*The DCU Ability project reflects the ethos of the university to do more than educate—we support, we facilitate development, we learn with our students, and we provide a person-centred approach that takes the whole person into consideration—their thoughts, their needs and their goals. It is important that the university population reflects the diversity of the national population without discrimination or prejudice*".

Results reflect a staff body who want to see the university's goals reached and see an inclusive and diverse student body as the way to achieve them. They want to see "*different perspectives, being good humans to one another*"; they would like the university to have "*awareness, a greater understanding and breaking down barriers*", and see the benefits for all by " . . . *learning from students with disabilities*". However, they also challenge the university: "*Discrimination in terms of disability is a form of oppression and it's about time that institutions such as DCU step up . . . It is about time*". "*The project goes beyond optics and tries to make a real change for the better. Improved publicity would help overcome the tight purse strings of DCU and make this an integrated part of university life*". Staff members clearly are supportive of the project, and see the university as having a significant role to play in

increasing inclusion and diversity while benefitting the entire University. " ... *the DCU community would benefit from the inclusivity of people with disabilities, increase understanding of the challenges faced by people with disabilities and by being part of the solution to those challenges. DCU brand would benefit by leading the way*".

## 7. Discussion

The results of this study contribute to making a case for inclusion while acknowledging the challenges of supporting a more diverse student body. However, it must be acknowledged that while rich data were obtained, particularly with the open-ended questions, a response rate of 4% is very low. Diversity and inclusion in education is not a new concept, and there are some recent studies that pay particular attention to the views of those working directly in educational institutions, including tutors, lecturers and other faculty members [12,34]. In general, those who engage in such research report positive educators' perceptions of inclusion [12,35–37]. However, a favourable climate is not always reported by the academic community. Other studies cite attitude barriers (perceived or otherwise) on the part of academic staff as a direct challenge to learners with disabilities [38–42]. Educators' views are important to make a case for inclusion, given the assumption that successful implementation of inclusive programs (leading to greater access and equality) depends on all teaching staff being positive [43,44].

Inclusion is at the heart of the DCU Ability project. It is understood as the process of increasing opportunities, participation and benefits to every learner who wishes to enter the educational system [45]. Over half of surveyed participants, 62% (*n* = 68) agreed or strongly agreed that the project allows DCU to be more inclusive to people with disabilities. Staff have elaborated on their experiences of inclusion. Nearly half, 43% (*n* = 47) of the sample strongly agreed that the DCU community had benefited from working closely with young adults with intellectual disabilities. A significant number of comments from participants show support for greater diversity within the university. These findings resonate with other studies, which documented the benefits of inclusion on an academic faculty [18,34,35,46]. Results also echo previous research undertaken in Trinity College Dublin by O'Connor and colleagues [20], who interviewed eleven lecturers in relation to their perceptions of people with intellectual disabilities who audited their classes. These lecturers cited benefits to this experience, including increased motivation, with opportunities to grow competence and reflect on their teaching. Furthermore, DCU employees reported a brighter outlook in their workplace. A total of 62% (*n* = 68) strongly agreed or agreed that awareness of the project led to positive attitudes about working in their university. The strength of the theme of diversity and inclusion in this study provides evidence that staff see the benefits and want a more inclusive university.

However, the significant challenge around communication shown in this study indicates that communication is key to successful outputs in higher education [47,48]. Results suggest some limited understanding and awareness of the DCU Ability project within the DCU staff community. This result was the least positive in the survey, as while many (34%—*n* = 37) did agree or strongly agree that they understood what the project was trying to achieve, many participants (66%—*n* = 73) ranked their understanding between 5 and 1. This result may be linked to the broader, deep-seated communication issues in university life, as reflected in some of the comments from participants.

With five separate campuses, DCU is home to over 17,000 students and nearly 1700 staff, excluding research personnel; it is not surprising that communication of one project within such a large organisation may be difficult. The results reflect the complexity of communicating the project's goals and initiatives with DCU employees across campuses. Reduced visibility of the DCU Ability project has arguably contributed to the broader communication issues present in most 21st-century universities [49–51]. Technology (including smartphones and emails) has changed the shape of higher education systems, as staff primarily use emails to communicate with one another. Given the increasing volumes of emails, smaller projects like this one may not receive adequate attention amongst staff on

the ground. Faculty staff report excess and unmanageable volumes of work emails [52–54], leading to information overload [55].

This study highlights the difficulty for the DCU Ability project to find its voice in a growing and dynamic university. However, despite these communication issues, an informed and united academic community is key to supporting students with intellectual disabilities [1]. Thus, more intensive efforts to engage with academic staff about the DCU Ability project beyond the standard open email channels are recommended.

University-level communication strategies must be strengthened on a macro level to allow more accessible dialogue and collaboration amongst a wide circle of scholars and students. It appears that email is the predominant form of communication for academic staff, so this is by no means an easy endeavour. In combination with open email channels, communication strategies for access projects must seek alternative routes to reach, and most importantly engage with, the scholarly community.

Recommendations from participants advocating for in-person or face-to-face events to promote the programme and receive educational training on inclusion is useful. Specifically, promotional and communication measures for the project should reflect a human touch (such as face-to-face or small online educational events) so as to connect with staff in their day-to-day lives on campus. In doing so, small projects can find their voice. In return, strong ties are forged within the disability and academic community, leading to greater participation and inclusion of disabled students in higher education.

The results of the survey revealed how inclusion programmes like the DCU Ability project are a cornerstone to DCU's Strategic Plan. Particularly, the offer of support to people with intellectual disabilities so they can experience university life and improve their employment prospects, aligns with fostering the potential of every student and preparing them for the 21st-century workforce.

Furthermore, mapping the alignment of inclusive programmes to the university's visions for the future made a case for additional initiatives. It also supported the argument for the inclusion of people with intellectual disabilities at third level. Results reveal that such projects to promote inclusion will be well received and promoted by individual staff within the university, reflecting the arguments made by Plotner and Marshall [56] that faculty staff support was a factor that played a significant role in provision of inclusive education opportunities for adults with intellectual disabilities. This was evidenced by the majority of staff (77.3%) who strongly agreed that students with disabilities should have equal access to their institution, and nearly half of the sample would recommend this programme to others. Findings highlight the important role academic staff play on the road to inclusive education. A recommendation therefore is that staff are given more opportunity to become involved at different stages of projects such as DCU Ability.

Recognising the role of diversity projects like the DCU Ability project and how they fit into the higher education landscape is key to making a case for the inclusion of people with intellectual disabilities, both in terms of their career prospects and their opportunities to attend university. However, students with disabilities encounter challenges to participate in colleges fully, often referred to in the literature as "ableist" educational environments [57].

While the focus of this study was on tracing the perspectives of the academic staff, research is required to build on the work of others who have investigated the student experience [6,58]. This knowledge is also essential to build "*connections between structural conditions and the lived reality of people*" [59] (p. 1), particularly in the context of higher education. Such work will also build on research reflecting the voices of students with intellectual disabilities.

## 8. Conclusions

This study provided important insights into the perspectives of academic staff working in a university that hosts an inclusive education and work initiative for young adults with intellectual disabilities. Understanding the perspectives of university staff is essential to an understanding of the broader context of inclusive education programmes, and the

insights gained from staff have highlighted important strengths and areas for growth for DCU Ability.

Findings from this study suggest that overall, participants had positive views about their experiences of the DCU Ability project. Acknowledging the limitations of a low response rate, University staff have highlighted issues in the area of communication and challenged the university to align projects like this more with strategic goals. These perspectives have proved useful in identifying areas for further development and growth of the DCU Ability project.

There is limited available evidence outlining the perspectives of university staff who are not directly involved in working with learners with intellectual disabilities. This study is useful in beginning to inform the broader context of one university-based programme in Ireland for learners with intellectual disabilities, but the findings may also be pertinent to similar programmes. A broader study which includes demographic and experience details of respondents would be useful to expand on these findings.

The study has also highlighted the need for additional research in order to understand the experiences both of learners with intellectual disabilities, and of family members who have experienced this unique programme, as well as those of service providers and other stakeholders who are involved in supporting learners to attend the course. Such research would inform evaluation processes and future planning for the programme.

**Author Contributions:** Conceptualization, D.C., T.A., S.R., E.K. and M.P.; methodology, D.C., T.A., E.K. and M.P.; formal analysis, D.C. and A.C.; original draft preparation, D.C., A.C., E.K., S.R. and M.P.; writing—review and editing, D.C.; visualization, D.C., T.A., S.R. and E.K.; project administration, T.A.; funding acquisition, D.C. All authors have read and agreed to the published version of the manuscript.

**Funding:** The DCU Ability project was one of 27 projects co-financed by the Irish Government and the European Social Fund as part of the ESF Programme for Employability, Inclusion and Learning 2014–2020. The project was extended to December 2022 with additional funding from the Irish Government with support from the Dormant Accounts Fund.

**Institutional Review Board Statement:** Ethical approval for this study was obtained from Dublin City University, Research Ethics Committee. The study was conducted in accordance with the Declaration of Helsinki, and approved by Ethics Committee of Dublin City University (DCUREC 2020/139 1 July 2020) for studies involving humans.

**Informed Consent Statement:** This was an anonymous online survey and by completing the survey participants consent was given to include their answers in our analysis. There were no identifying marks on replies and no names can be used in any reports or publications. Therefore informed consent was obtained from all subjects involved in the study.

**Data Availability Statement:** Data supporting reported results can be obtained from the principle investigator Deirdre Corby.

**Acknowledgments:** Authors would like to thank DCU staff who completed the survey.

**Conflicts of Interest:** The authors declare no conflict of interest. The funders had no role in the design of the study; in the collection, analyses, or interpretation of data; in the writing of the manuscript, or in the decision to publish the results.

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
