# Peer review of "Making a Case for the Inclusion of People with Intellectual Disabilities in Higher Education"

_disabilities, doi:10.3390/disabilities2030029_

Round 1

Reviewer 1 Report

Thank you for the opportunity to review the manuscript, "Making a Case for the Inclusion of People with Intellectual Disabilities in Higher Education." I really enjoyed reading this paper and felt that the authors did a tremendous job describing inclusive higher education and its benefits. I believe this paper will be suitable for publication, pending a few minor revisions.

First, although I recognize that this article focuses on inclusion in the higher ed setting, I think it would be appropriate to provide some context for current inclusive practices in Ireland for younger students with ID (primary and secondary school). Are students with ID currently included in general education classes in primary and secondary schools? I think it is important to contextualize the university programs within the larger education system and it's practices.

Second, you had an extremely small sample (due to a very low response rate to your survey). I think this needs to be acknowledged in your Discussion. Further, I think it would be beneficial to provide some data in regards to your sample - who were the respondents? What departments are they from? How long have they been at the university, etc.?

With the additions of these elements, I believe the manuscript will make a fine contribution to Disabilities and the growing literature in inclusive postsecondary education for students with ID.

Author Response

Dear Reviewer

Thank you for taking the time to review this paper and for your generous comments.

I will address them one by one here, and changes in the document have been highlighted in red and in italics here.

First, although I recognize that this article focuses on inclusion in the higher ed setting, I think it would be appropriate to provide some context for current inclusive practices in Ireland for younger learners with ID (primary and secondary school). Are learners with ID currently included in general education classes in primary and secondary schools? I think it is important to contextualize the university programs within the larger education system and it's practices.

This has been added to the introduction: In the Republic of Ireland children who have special needs attend either mainstream school with supports or special classes in mainstream schools, or attend special schools. Once their primary and secondary education is complete at the age of eighteen, those with intellectual disabilities tend to progress to specialist services rather than higher education.

Second, you had an extremely small sample (due to a very low response rate to your survey). I think this needs to be acknowledged in your Discussion. Further, I think it would be beneficial to provide some data in regards to your sample - who were the respondents? What departments are they from? How long have they been at the university, etc.?

This is a very reasonable point and also made by reviewer 2. As we didn’t collect any data on respondents this is a limitation and has been acknowledged a few times in the paper now by adding the following.

In section 5: Materials and Methods I have added

Therefore a wide range of academic, technical and professional staff would have received the email with the survey link. It is most likely that the majority of those who responded to the survey had direct experience of working with the learners. …. Participation was anonymous, with no demographic or identifying information collected.

In Discussion:

However it must be acknowledged that while rich data was obtained, particularly with the open ended questions, a response rate of 4% is very low.

In the Conclusion:

Acknowledging the limitations of a low response rate University staff have highlighted .....

A broader study which includes demographic and experience details of respondents would be useful to expand on these findings.

Reviewer 3 Report

I found this a personally interesting paper as I had not previously seen much empirical research about such projects focussed on intellectual disabilities in HE. The paper sets the DCU Ability project in an international context with a range of sources about related projects in HE in other countries. There is some background to the context of this project - its time limited funding etc, but not key points about how it runs and where it is located in the university as regards the formal academic courses and specifically how it was set up. I think this is relevant to one of the main findings which is about communication within the university about this project. The study of staff perspectives was basically done well, but more could have been written about who the respondents were, especially as they were about 4% of the staff. The significance of this low response rate for whose views are represented here are crucial to any overall assessment of what the study is reporting. On a more technical matter, the chart of responses could give a legend about what the different colour bars represent. I think I know but this be be made clear. 

I think this paper deserves to be published but with some minor revisions:

1. insert legend for response chart

2. more detail about the context of this project - specifics about its aims and how they relate to formal academic courses, links between students on those courses and those involved in this project, how it was set up and negotiated/communicated to staff at inception, what level of intellectual; abilities/disabilities were recruited for this project and so on.

3. demographic details of those who responded to the questionnaire so the results can be presented in terms whose views are being represented. 

Author Response

Dear Reviewer

Thank you for taking the time to review this paper and for your generous comments.

I will address them one by one here, and changes in the document have been highlighted in red and in italics here.

1. Insert legend for response chart

Legend has been inserted and chart updated

  1. more detail about the context of this project - specifics about its aims and how they relate to formal academic courses, links between learners on those courses and those involved in this project, how it was set up and negotiated/communicated to staff at inception, what level of intellectual; abilities/disabilities were recruited for this project and so on.

This has been added to the introduction

Based in the School of Nursing, Psychotherapy and Community Health, a School within the DCU Faculty of Science and Health it aims to support people with intellectual disabilities in building the skills and confidence they need to progress to further education or towards the workplace. Learners were recruited through services for people with disabilities and initially open evenings were held to inform learners, families and service staff about the project. Potential learners would meet with the Ability team to ensure the course would meet their needs and help them move towards employment.

  1. demographic details of those who responded to the questionnaire so the results can be presented in terms whose views are being represented. 

This is a very reasonable point and also made by reviewer 1. As we didn’t collect any data on respondents this is a limitation and has been acknowledged a few times in the paper now by adding the following.

In section 5: Materials and Methods I have added

Therefore a wide range of academic, technical and professional staff would have received the email with the survey link. It is most likely that the majority of those who responded to the survey had direct experience of working with the learners. …. Participation was anonymous, with no demographic or identifying information collected.

In Discussion:

However it must be acknowledged that while rich data was obtained, particularly with the open ended questions, a response rate of 4% is very low.

In the Conclusion:

Acknowledging the limitations of a low response rate University staff have highlighted .....

A broader study which includes demographic and experience details of respondents would be useful to expand on these findings.